# Multi-task Learning for Aggregated Data using Gaussian Processes

**Fariba Yousefi**      **Michael Thomas Smith**      **Mauricio A. Álvarez**

Department of Computer Science, University of Sheffield
{f.yousefi, m.t.smith, mauricio.alvarez}@sheffield.ac.uk

## Abstract

Aggregated data is commonplace in areas such as epidemiology and demography. For example, census data for a population is usually given as averages defined over time periods or spatial resolutions (cities, regions or countries). In this paper, we present a novel multi-task learning model based on Gaussian processes for joint learning of variables that have been aggregated at different input scales. Our model represents each task as the linear combination of the realizations of latent processes that are integrated at a different scale per task. We are then able to compute the cross-covariance between the different tasks either analytically or numerically. We also allow each task to have a potentially different likelihood model and provide a variational lower bound that can be optimised in a stochastic fashion making our model suitable for larger datasets. We show examples of the model in a synthetic example, a fertility dataset and an air pollution prediction application.

## 1   Introduction

Many datasets in fields like ecology, epidemiology, remote sensing, sensor networks and demography appear naturally aggregated, that is, variables in these datasets are measured or collected in intervals, areas or supports of different shapes and sizes. For example, census data are usually sampled or collected as aggregated at different administrative divisions, e.g. borough, town, postcode or city levels. In sensor networks, correlated variables are measured at different resolutions or scales. In the near future, air pollution monitoring across cities and regions could be done using a combination of a few high-quality low time-resolution sensors and several low-quality (low-cost) high time-resolution sensors. Joint modelling of the variables registered in the census data or the variables measured using different sensor configurations at different scales can improve predictions at the point or support levels.

In this paper, we are interested in providing a general framework for multi-task learning on these types of datasets. Our motivation is to use multi-task learning to jointly learn models for different tasks where each task is defined at (potentially) a different support of any shape and size and has a (potentially) different nature, i.e. it is a continuous, binary, categorical or count variable. We appeal to the flexibility of Gaussian processes (GPs) for developing a prior over such type of datasets and we also provide a scalable approach for variational Bayesian inference.

Gaussian processes have been used before for aggregated data [Smith et al., Law et al., 2018, Tanaka et al., 2019] and also in the context of the related field of *multiple instance learning* [Kim and De la Torre, 2010, Kandemir et al., 2016, Haußmann et al., 2017]. In multiple instance learning, each instance in the dataset consists of a set (or *bag*) of inputs with only one output (or label) for that whole set. The aim is to provide predictions at the level of individual inputs. Smith et al. provide a new kernel function to handle single regression tasks defined at different supports. They use

cross-validation for hyperparameter selection. Law et al. [2018] use the weighted sum of a latent function evaluated at different inputs as the prior for the rate of a Poisson likelihood. The latent function follows a GP prior. The authors use stochastic variational inference (SVI) for approximating the posterior distributions. Tanaka et al. [2019] mainly use GPs for creating data from different auxiliary sources. Furthermore, they only consider Gaussian regression and they do not include inducing variables. While Smith et al. and Law et al. [2018] perform the aggregation at the latent prior stage, Kim and De la Torre [2010], Kandemir et al. [2016] and Haußmann et al. [2017] perform the aggregation at the likelihood level. These three approaches target a binary classification problem. Both Kim and De la Torre [2010] and Haußmann et al. [2017] focus on the case for which the label of the bag corresponds to the maximum of the (unobserved) individual labels of each input. Kim and De la Torre [2010] approximate the maximum using a softmax function computed using a latent GP prior evaluated across the individual elements of the bag. They use the Laplace approximation for computing the approximated posterior [Rasmussen and Williams, 2006]. Haußmann et al. [2017], on the other hand, approximate the maximum using the so called *bag label likelihood*, introduced by the authors, which is similar to a Bernoulli likelihood with soft labels given by a convex combination between the bag labels and the maximum of the (latent) individual labels. The latent individual labels in turn follow Bernoulli likelihoods with parameters given by a GP. The authors provide a variational bound and include inducing inputs for scalable Bayesian inference. Kandemir et al. [2016] follow a similar approach to Law et al. [2018] equivalent to setting all the weights in Law et al.'s model to one. The sum is then used to modulate the parameter of a Bernoulli likelihood that models the bag labels. They use a Fully Independent Training Conditional approximation for the latent GP prior [Snelson and Ghahramani, 2006]. In contrast to these previous works, we provide a multi-task learning model for aggregated data that scales to large datasets and allows for heterogeneous outputs. At the time of submission of this paper, the idea of using multi-task learning for aggregated datasets was simultaneously proposed by Hamelijnck et al. and Tanaka et al., two additional models to the one we propose in this paper. In our work, we allow heterogenous likelihoods which is different to both Hamelijnck et al. and Tanaka et al.. We also allow an exact solution to the integration of the latent function through the kernel in Smith et al., which is different to Hamelijnck et al.. Also, for computational complexity, inducing inputs are used, another difference from the work in Tanaka et al.. Other relevant work is described in Section 3.

For building the multi-task learning model we appeal to the linear model of coregionalisation [Journel and Huijbregts, 1978, Goovaerts, 1997] that has gained popularity in the multi-task GP literature in recent years [Bonilla et al., 2008, Alvarez et al., 2012]. We also allow different likelihood functions [Moreno-Muñoz et al., 2018] and different input supports per individual task. Moreover, we introduce inducing inputs at the level of the underlying common set of latent functions, an idea initially proposed in Alvarez and Lawrence [2009]. We then use stochastic variational inference for GPs [Hensman et al., 2013] leading to an approximation similar to the one obtained in Moreno-Muñoz et al. [2018]. Empirical results show that the multi-task learning approach developed here provides accurate predictions in different challenging datasets where tasks have different supports.

## 2 Multi-task learning for aggregated data at different scales

In this section we first define the basic model in the single-task setting. We then extend the model to the multi-task setting and finally provide details for the stochastic variational formulation for approximate Bayesian inference.

### 2.1 Change of support using Gaussian processes

Change of support has been studied in geostatistics before [Gotway and Young, 2002]. In this paper, we use a formulation similar to Kyriakidis [2004]. We start by defining a stochastic process over the input interval $(x_a, x_b)$ using

$$f(x_a, x_b) = \frac{1}{\Delta_x} \int_{x_a}^{x_b} u(z) dz,$$

where $u(z)$ is a latent stochastic process that we assume follows a Gaussian process with zero mean and covariance $k(z, z')$ and $\Delta_x = |x_b - x_a|$. Dividing by $\Delta_x$ helps to keep the proportionality between the length of the interval and the area under $u(z)$ in the interval. In other words, the process $f(\cdot)$ is modeled as a density meaning that inputs with widely differing supports will

behave in a similar way. The first two moments for $f(x_a, x_b)$ are given as $\mathbb{E}[f(x_a, x_b)] = 0$ and $\mathbb{E}[f(x_a, x_b), f(x'_a, x'_b)] = \frac{1}{\Delta_x \Delta_{x'}} \int_{x_a}^{x_b} \int_{x'_a}^{x'_b} \mathbb{E}[u(z)u(z')]dz'dz$. The covariance for $f(x_a, x_b)$ follows as $\mathrm{cov}[f(x_a, x_b), f(x'_a, x'_b)] = \frac{1}{\Delta_x \Delta_{x'}} \int_{x_a}^{x_b} \int_{x'_a}^{x'_b} k(z, z')dz'dz$ since $\mathbb{E}[u(z)] = 0$. Let us use $k(x_a, x_b, x'_a, x'_b)$ to refer to $\mathrm{cov}[f(x_a, x_b), f(x'_a, x'_b)]$. We can now use these mean and covariance functions for representing the Gaussian process prior for $f(x_a, x_b) \sim \mathcal{GP}(0, k(x_a, x_b, x'_a, x'_b))$.

For some forms of $k(z, z')$ it is possible to obtain an analytical expression for $k(x_a, x_b, x'_a, x'_b)$. For example, if $k(z, z')$ follows an Exponentiated-Quadratic (EQ) covariance form, $k(z, z') = \sigma^2 \exp\{-\frac{(z-z')^2}{\ell^2}\}$, where $\sigma^2$ is the variance of the kernel and $\ell$ is the length-scale, it can be shown that $k(x_a, x_b, x'_a, x'_b)$ follows as

$$k(x_a, x_b, x'_a, x'_b) = \frac{\sigma^2 \ell^2}{2\Delta_x \Delta_{x'}} \left[ h\left(\frac{x_b - x'_a}{\ell}\right) + h\left(\frac{x_a - x'_b}{\ell}\right) - h\left(\frac{x_a - x'_a}{\ell}\right) - h\left(\frac{x_b - x'_b}{\ell}\right) \right],$$

where $h(z) = \sqrt{\pi} z \, \mathrm{erf}(z) + e^{-z^2}$ with $\mathrm{erf}(z)$, the error function defined as $\mathrm{erf}(z) = \frac{2}{\sqrt{\pi}} \int_0^z e^{-r^2} dr$. Other kernels for $k(z, z')$ also lead to analytical expressions for $k(x_a, x_b, x'_a, x'_b)$. See for example Smith et al..

So far, we have restricted the exposition to one-dimensional intervals. However, we can define the stochastic process $f$ over a general support $v$, with measure $|v|$, using

$$f(v) = \frac{1}{|v|} \int_{\mathbf{z} \in v} u(\mathbf{z}) d\mathbf{z}.$$

The support $v$ generally refers to an area or volume of any shape or size. Following similar assumptions to the ones we used for $f(x_a, x_b)$, we can build a GP prior to represent $f(v)$ with covariance $k(v, v')$ defined as $k(v, v') = \frac{1}{|v||v'|} \int_{\mathbf{z} \in v} \int_{\mathbf{z}' \in v'} k(\mathbf{z}, \mathbf{z}')d\mathbf{z}'d\mathbf{z}$. Let $\mathbf{z} \in \mathbb{R}^p$. If the support $v$ has a regular shape, e.g. a hyperrectangle, then assumptions on $u(\mathbf{z})$ such as additivity or factorization across input dimensions will lead to kernels that can be expressed as addition of kernels or product of kernels acting over a single dimension. For example, let $u(\mathbf{z}) = \prod_{i=1}^p u_i(z_i)$, where $\mathbf{z} = [z_1, \cdots, z_p]^\top$, and a GP over each $u_i(z_i) \sim \mathcal{GP}(0, k(z_i, z'_i))$. If each $k(z_i, z'_i)$ is an EQ kernel, then $k(v, v') = \prod_{i=1}^p k(x_{i,a}, x_{i,b}, x'_{i,a}, x'_{i,b})$, where $(x_{i,a}, x_{i,b})$ and $(x'_{i,a}, x'_{i,b})$ are the intervals across each input dimension. If the support $v$ does not follow a regular shape, i.e it is a polytope, then we can approximate the double integration by numerical integration inside the support.

## 2.2 Multi-task learning setting

Our inspiration for multi-task learning is the linear model of coregionalisation (LMC) [Journel and Huijbregts, 1978]. This model has connections with other multi-task learning approaches that use kernel methods. For example, Teh et al. [2005] and Bonilla et al. [2008] are particular cases of LMC. A detailed review is available in Alvarez et al. [2012]. In the LMC, each output (or task in our case) is represented as a linear combination of a common set of latent Gaussian processes. Let $\{u_q(\mathbf{z})\}_{q=1}^Q$ be a set of $Q$ GPs with zero means and covariance functions $k_q(\mathbf{z}, \mathbf{z}')$. Each GP $u_q(\mathbf{z})$ is sampled independently and identically $R_q$ times to produce $\{u_q^i(\mathbf{z})\}_{i=1, q=1}^{R_q, Q}$ realizations that are used to represent the outputs. Let $\{f_d(v)\}_{d=1}^D$ be a set of tasks where each task is defined at a different support $v$. We use the set of realizations $u_q^i(\mathbf{z})$ to represent each task $f_d(v)$ as

$$f_d(v) = \sum_{q=1}^Q \sum_{i=1}^{R_q} \frac{a_{d,q}^i}{|v|} \int_{\mathbf{z} \in v} u_q^i(\mathbf{z}) d\mathbf{z}, \tag{1}$$

where the coefficients $a_{d,q}^i$ weight the contribution of each integral term to $f_d(v)$. Since $\mathrm{cov}[u_q^i(\mathbf{z}), u_{q'}^{i'}(\mathbf{z}')] = k_q(\mathbf{z}, \mathbf{z}')\delta_{q,q'}\delta_{i,i'}$, with $\delta_{\alpha,\beta}$ the Kronecker delta between $\alpha$ and $\beta$, the cross-covariance $k_{f_d, f_{d'}}(v, v')$ between $f_d(v)$ and $f_{d'}(v')$ is then given as

$$k_{f_d, f_{d'}}(v, v') = \sum_{q=1}^Q \frac{b_{d,d'}^q}{|v||v'|} \int_{\mathbf{z} \in v} \int_{\mathbf{z}' \in v'} k_q(\mathbf{z}, \mathbf{z}')d\mathbf{z}'d\mathbf{z},$$

where $b_{d,d'}^q = \sum_{i=1}^{R_q} a_{d,q}^i a_{d',q}^i$. Following the discussion in Section 2.1, the double integral can be solved analytically for some options of $\upsilon$, $\upsilon'$ and $k_q(\mathbf{z}, \mathbf{z}')$. Generally, a numerical approximation can be obtained.

It is also worth mentioning at this point that the model does not require all the tasks to be defined at the area level. Some of the tasks could also be defined at the point level. Say for example that $f_d$ is defined at the support level $\upsilon$, $f_d(\upsilon)$, whereas $f_{d'}$ is defined at the point level, say $\mathbf{x} \in \mathbb{R}^p$, $f_{d'}(\mathbf{x})$. In this case, $f_{d'}(\mathbf{x}) = \sum_{q=1}^{Q} \sum_{i=1}^{R_q} a_{d',q}^i u_q^i(\mathbf{x})$. We can still compute the cross-covariance between $f_d(\upsilon)$ and $f_{d'}(\mathbf{x})$, $k_{f_d,f_{d'}}(\upsilon, \mathbf{x})$, leading to, $k_{f_d,f_{d'}}(\upsilon, \mathbf{x}) = \sum_{q=1}^{Q} \frac{b_{d,d'}^q}{|\upsilon|} \int_{\mathbf{z} \in \upsilon} k_q(\mathbf{z}, \mathbf{x}) d\mathbf{z}$. For the case $Q = 1$ and $p = 1$ (i.e. dimensionality of the input space), this is, $z, z', x \in \mathbb{R}$, $\upsilon = (x_a, x_b)$ and an EQ kernel for $k(z, z')$, we get $k_{f_d,f_{d'}}(\upsilon, x) = \frac{b_{d,d'}}{\Delta_x} \int_{x_a}^{x_b} k(z, x) dz = \frac{b_{d,d'}\ell}{2\Delta_x} \left[ \mathrm{erf}\left(\frac{x_b - x}{\ell}\right) + \mathrm{erf}\left(\frac{x - x_a}{\ell}\right) \right]$ (we used $\sigma^2 = 1$ to avoid an overparameterization for the variance). Again, if $\upsilon$ does not have a regular shape, we can approximate the integral numerically.

Let us define the vector-valued function $\mathbf{f}(\upsilon) = [f_1(\upsilon), \cdots, f_D(\upsilon)]^\top$. A GP prior over $\mathbf{f}(\upsilon)$ can use the kernel defined above so that

$$\mathbf{f}(\upsilon) \sim \mathcal{GP}\left(\mathbf{0}, \sum_{q=1}^{Q} \frac{1}{|\upsilon||\upsilon'|} \mathbf{B}_q \int_{\mathbf{z} \in \upsilon} \int_{\mathbf{z}' \in \upsilon'} k_q(\mathbf{z}, \mathbf{z}') d\mathbf{z}' d\mathbf{z}\right),$$

where each $\mathbf{B}_q \in \mathbb{R}^{D \times D}$ is known as a coregionalisation matrix. The scalar term $\int_{\mathbf{z} \in \upsilon} \int_{\mathbf{z}' \in \upsilon'} k_q(\mathbf{z}, \mathbf{z}') d\mathbf{z}' d\mathbf{z}$ modulates $\mathbf{B}_q$ as a function of $\upsilon$ and $\upsilon'$.

The prior above can be used for modulating the parameters of likelihood functions that model the observed data. The most simple case corresponds to the multi-task regression problem that can be modeled using a multivariate Gaussian distribution. Let $\mathbf{y}(\upsilon) = [y_1(\upsilon), \cdots, y_D(\upsilon)]^\top$ be a random vector modeling the observed data as a function of $\upsilon$. In the multi-task regression problem $\mathbf{y}(\upsilon) \sim \mathcal{N}(\boldsymbol{\mu}(\upsilon), \boldsymbol{\Sigma})$, where $\boldsymbol{\mu}(\upsilon) = [\mu_1(\upsilon), \cdots, \mu_D(\upsilon)]^\top$ is the mean vector and $\boldsymbol{\Sigma}$ is a diagonal matrix with entries $\{\sigma_{y_d}^2\}_{d=1}^D$. We can use the GP prior $\mathbf{f}(\upsilon)$ as the prior over the mean vector $\boldsymbol{\mu}(\upsilon) \sim \mathbf{f}(\upsilon)$. Since both the likelihood and the prior are Gaussian, both the marginal distribution for $\mathbf{y}(\upsilon)$ and the posterior distribution of $\mathbf{f}(\upsilon)$ given $\mathbf{y}(\upsilon)$ can be computed analytically. For example, the marginal distribution for $\mathbf{y}(\upsilon)$ is given as $\mathbf{y}(\upsilon) \sim \mathcal{N}(\mathbf{0}, \sum_{q=1}^{Q} \frac{1}{|\upsilon||\upsilon'|} \mathbf{B}_q \int_{\mathbf{z} \in \upsilon} \int_{\mathbf{z}' \in \upsilon'} k_q(\mathbf{z}, \mathbf{z}') d\mathbf{z}' d\mathbf{z} + \boldsymbol{\Sigma})$. Moreno-Muñoz et al. [2018] introduced the idea of allowing each task to have a different likelihood function and modulated the parameters of that likelihood function using one or more elements in the vector-valued GP prior. For that general case, the marginal likelihood and the posterior distribution cannot be computed in closed form.

## 2.3 Stochastic variational inference

Let $\mathcal{D} = \{\boldsymbol{\Upsilon}, \mathbf{y}\}$ be a dataset of multiple tasks with potentially different supports per task, where $\boldsymbol{\Upsilon} = \{\boldsymbol{\upsilon}_d\}_{d=1}^D$, with $\boldsymbol{\upsilon}_d = [\upsilon_{d,1}, \cdots, \upsilon_{d,N_d}]^\top$, and $\mathbf{y} = [\mathbf{y}_1, \cdots, \mathbf{y}_D]^\top$, with $\mathbf{y}_d = [y_{d,1}, \cdots, y_{d,N_d}]^\top$ and $y_{d,j} = y_d(\upsilon_{d,j})$. Notice that $\mathbf{y}$ without $\upsilon$ refers to the output vector for the dataset. We are interested in computing the posterior distribution $p(\mathbf{f}|\mathbf{y}) = p(\mathbf{y}|\mathbf{f})p(\mathbf{f})/p(\mathbf{y})$, where $\mathbf{f} = [\mathbf{f}_1, \cdots, \mathbf{f}_D]^\top$, with $\mathbf{f}_d = [f_{d,1}, \cdots, f_{d,N_d}]^\top$ and $f_{d,j} = f_d(\upsilon_{d,j})$. In this paper, we will use stochastic variational inference to compute a deterministic approximation of the posterior distribution $p(\mathbf{f}|\mathbf{y}) \approx q(\mathbf{f})$, by means of the the well known idea of *inducing variables*. In contrast to the use of SVI for traditional Gaussian processes, where the inducing variables are defined at the level of the process $\mathbf{f}$, we follow Álvarez et al. [2010] and Moreno-Muñoz et al. [2018], and define the inducing variables at the level of the latent processes $u_q(\mathbf{z})$. For simplicity in the notation, we assume $R_q = 1$. Let $\mathbf{u} = \{\mathbf{u}_q\}_{q=1}^Q$ be the set of inducing variables, where $\mathbf{u}_q = [u_q(\mathbf{z}_1), \cdots, u_q(\mathbf{z}_M)]^\top$, with $\mathbf{Z} = \{\mathbf{z}_m\}_{m=1}^M$ the inducing inputs. Notice also that we have used a common set of inducing inputs $\mathbf{Z}$ for all latent functions but we can easily define a set $\mathbf{Z}_q$ per inducing vector $\mathbf{u}_q$.

For the multi-task regression case, it is possible to compute an analytical expression for the Gaussian posterior distribution over the inducing variables $\mathbf{u}$, $q(\mathbf{u})$, following a similar approach to Álvarez et al. [2010]. However, such approximation is only valid for the case in which the likelihood model $p(\mathbf{y}|\mathbf{f})$ is Gaussian and the variational bound obtained is not amenable for stochastic optimisation. An alternative for finding $q(\mathbf{u})$ also establishes a lower-bound for the log-marginal likelihood $\log p(\mathbf{y})$,

but uses numerical optimisation for maximising the bound with respect to the mean parameters, $\boldsymbol{\mu}$, and the covariance parameters, $\mathbf{S}$, for the Gaussian distribution $q(\mathbf{u}) \sim \mathcal{N}(\boldsymbol{\mu}, \mathbf{S})$ [Moreno-Muñoz et al., 2018]. Such numerical procedure can be used for any likelihood model $p(\mathbf{y}|\mathbf{f})$ and the optimisation can be performed using mini-batches. We follow this approach.

**Lower-bound**    The lower bound for the log-marginal likelihood follows as

$$\log p(\mathbf{y}) \geq \int \int q(\mathbf{f}, \mathbf{u}) \log \frac{p(\mathbf{y}|\mathbf{f})p(\mathbf{f}|\mathbf{u})p(\mathbf{u})}{q(\mathbf{f}, \mathbf{u})} d\mathbf{f} d\mathbf{u} = \mathcal{L},$$

where $q(\mathbf{f}, \mathbf{u}) = p(\mathbf{f}|\mathbf{u})q(\mathbf{u})$, $p(\mathbf{f}|\mathbf{u}) \sim \mathcal{N}(\mathbf{K_{fu}}\mathbf{K_{uu}^{-1}}\mathbf{u}, \mathbf{K_{ff}} - \mathbf{K_{fu}}\mathbf{K_{uu}^{-1}}\mathbf{K_{fu}^{\top}})$, and $p(\mathbf{u}) \sim \mathcal{N}(\mathbf{0}, \mathbf{K_{uu}})$ is the prior over the inducing variables. Here $\mathbf{K_{fu}}$ is a blockwise matrix with matrices $\mathbf{K_{f_d, u_q}}$. In turn each of these matrices have entries given by $k_{f_d, u_q}(v, \mathbf{z}') = \frac{a_{d,q}}{|v|} \int_{\mathbf{z} \in v} k_q(\mathbf{z}, \mathbf{z}') d\mathbf{z}$. Similarly, $\mathbf{K_{uu}}$ is a block-diagonal matrix with blocks given by $\mathbf{K}_q$ with entries computed using $k_q(\mathbf{z}, \mathbf{z}')$. The optimal $q(\mathbf{u})$ is chosen by numerically maximizing $\mathcal{L}$ with respect to the parameters $\boldsymbol{\mu}$ and $\mathbf{S}$. To ensure a valid covariance matrix $\mathbf{S}$ we optimise the Cholesky factor $\mathbf{L}$ for $\mathbf{S} = \mathbf{LL}^{\top}$. See Appendix A.1 for more details on the lower bound. The computational complexity is similar to the one for the model in Moreno-Muñoz et al. [2018], $\mathcal{O}(QM^3 + JNQM^2)$, where $J$ depends on the types of likelihoods used for the different tasks. For example, if we model all the outputs using Gaussian likelihoods, then $J = D$. For details, see Moreno-Muñoz et al. [2018].

**Hyperparameter learning**    When using the multi-task learning method, we need to optimise the hyperparameters associated with the LMC, these are: the coregionalisation matrices $\mathbf{B}_q$, the hyperparameters of the kernels $k_q(\mathbf{z}, \mathbf{z}')$, and any other hyperparameter associated to the likelihood functions $p(\mathbf{y}|\mathbf{f})$ that has not been considered as a member of the latent vector $\mathbf{f}(v)$. Hyperparameter optimisation is done using the lower bound $\mathcal{L}$ as the objective function. First $\mathcal{L}$ is maximised with respect to the variational distribution $q(\mathbf{u})$ and then with respect to the hyperparameters. The two-steps are repeated one after the other until reaching convergence. Such style of optimisation is known as variational EM (Expectation-Maximization) when using the full dataset [Beal, 2003] or stochastic version, when employing mini-batches [Hoffman et al., 2013]. In the Expectation step we compute a variational posterior distribution and in the Maximization step we use a variational lower bound to find point estimates of any hyperparameters. For optimising the hyperparameters in $\mathbf{B}_q$, we also use a Cholesky decomposition for each matrix to ensure positive definiteness. So instead of optimising $\mathbf{B}_q$ directly, we optimise $\mathbf{L}_q$, where $\mathbf{B}_q = \mathbf{L}_q \mathbf{L}_q^{\top}$. For the experimental section, we use the EQ kernel for $k_q(\mathbf{z}, \mathbf{z})$, so we fix the variance for $k_q(\mathbf{z}, \mathbf{z})$ to one (the variance per output is already contained in the matrices $\mathbf{B}_q$) and optimise the length-scales $\ell_q$.

**Predictive distribution**    Given a new set of test inputs $\boldsymbol{\Upsilon}_*$, the predictive distribution for $p(\mathbf{y}_*|\mathbf{y}, \boldsymbol{\Upsilon}_*)$ is computed using $p(\mathbf{y}_*|\mathbf{y}, \boldsymbol{\Upsilon}_*) = \int_{\mathbf{f}_*} p(\mathbf{y}_*|\mathbf{f}_*) q(\mathbf{f}_*) d\mathbf{f}_*$, where $\mathbf{y}_*$ and $\mathbf{f}_*$ refer to the vector-valued functions $\mathbf{y}$ and $\mathbf{f}$ evaluated at $\boldsymbol{\Upsilon}_*$. Notice that $q(\mathbf{f}_*) \approx p(\mathbf{f}_*|\mathbf{y})$. Even though $\mathbf{y}$ does not appear explicitly in the expression for $q(\mathbf{f}_*)$, it has been used to compute the posterior for $q(\mathbf{u})$ through the optimisation of $\mathcal{L}$ where $\mathbf{y}$ is explicitly taken into account. We are usually interested in the mean prediction $\mathbb{E}[\mathbf{y}_*]$ and the predictive variance $\text{var}[\mathbf{y}_*]$. Both can be computed by exchanging integrals in the double integration over $\mathbf{y}_*$ and $\mathbf{f}_*$. See Appendix A.1 for more details on this.

# 3    Related work

Machine learning methods for different forms of aggregated datasets are also known under the names of *multiple instance learning*, *learning from label proportions* or *weakly supervised learning on aggregate outputs* [Kück and de Freitas, 2005, Musicant et al., 2007, Quadrianto et al., 2009, Patrini et al., 2014, Kotzias et al., 2015, Bhowmik et al., 2015]. Law et al. [2018] provided a summary of these different approaches. Typically these methods start with the following setting: each instance in the dataset is in the form of a set of inputs for which there is only one corresponding output (e.g. the proportion of class labels, an average or a sample statistic). The prediction problem usually consists then in predicting the individual outputs for the individual inputs in the set. The setting we present in this paper is slightly different in the sense that, in general, for each instance, the input corresponds to a support of any shape and size and the output corresponds to a vector-valued output. Moreover, each

task can have its own support. Similarly, while most of these ML approaches have been developed for either regression or classification, our model is built on top of Moreno-Muñoz et al. [2018], allowing each task to have a potentially different likelihood.

As mentioned in the introduction, Gaussian processes have also been used for multiple instance learning or aggregated data [Kim and De la Torre, 2010, Kandemir et al., 2016, Haußmann et al., 2017, Smith et al., Law et al., 2018, Tanaka et al., 2019, Hamelijnck et al., Tanaka et al.]. Compared to most of these previous approaches, our model goes beyond the single task problem and allows learning multiple tasks simultaneously. Each task can have its own support at training and test time. Compared to other multi-task approaches, we allow for heterogeneous outputs. Although our model was formulated for a continuous support $\mathbf{x} \in \upsilon_{d,j}$, we can also define it in terms of a finite set of (previously defined) inputs in the support, e.g. a set $\{\mathbf{x}_{d,j,1}, \cdots, \mathbf{x}_{d,j,K_{d,j}}\} \in \upsilon_{d,j}$ which is more akin to the bag formulations in these previous works. This would require changing the integral $\frac{1}{|\upsilon_{d,j}|} \int_{\mathbf{z} \in \upsilon_{d,j}} u_q^i(\mathbf{z})d\mathbf{z}$ in (1) for the sum $\frac{1}{K_{d,j}} \sum_{\forall \mathbf{x} \in \upsilon_{d,j}} u_q^i(\mathbf{x}_{d,j,k})$.

In geostatistics, a similar problem has been studied under the names of *downscaling* or *spatial disaggregation* [Zhang et al., 2014], particularly using different forms of *kriging* [Goovaerts, 1997]. It is also closely related to the problem of *change of support* described with detail in Gotway and Young [2002]. Block-to-point kriging (or area-to-point kriging if the support is defined in a surface) is a common method for downscaling, this is, provide predictions at the point level provided data at the block level [Kyriakidis, 2004, Goovaerts, 2010]. We extend the approach introduced in Kyriakidis [2004] later revisited by Goovaerts [2010] for count data, to the multi-task setting, including also a stochastic variational EM algorithm for scalable inference.

If we consider the high-resolution outputs as high-fidelity outputs and low-resolution outputs as low-fidelity outputs, our work also falls under the umbrella of *multi-fidelity models* where co-kriging using the linear model of coregionalisation has also been used as an alternative [Peherstorfer et al., 2018, Fernández-Godino et al.].

## 4   Experiments

In this section, we apply the multi-task learning model for prediction in three different datasets: a synthetic example for two tasks that each have a Poisson likelihood, a two-dimensional input dataset of fertility rates aggregated by year of conception and ages in Canada, and an air-pollution sensor network where one task corresponds to a high-accuracy, low-frequency particulate matter sensor and another task corresponds to a low-cost, low-accuracy, high resolution sensor. In these examples, we use $k$-means clustering over the input data, with $k = M$, to initialise the values of the inducing inputs, $\mathbf{Z}$, which are also kept fixed during optimisation. We assume the inducing inputs are points, but they could be defined as intervals or supports. For standard optimisation we used the LBFGS-B algorithm and when SVI was needed, the Adam optimiser, included in *climin* library, was used for the optimisation of the variational distribution (variational E-step) and the hyperparameters (variational M-step). The implementation is based on the GPy framework and is available on Github: `https://github.com/frb-yousefi/aggregated-multitask-gp`.

**Synthetic data**   In this section we evaluated our model with a synthetic dataset. For all of the experiments we use $Q = 1$ with an EQ covariance for the latent function $u_1(z)$. We set up a toy problem with $D = 2$ tasks, where both likelihood functions are Poisson. We sample from the latent vector-valued GP and use those samples to modulate the Poisson likelihoods using $\exp(f_1(\cdot))$ and $\exp(f_2(\cdot))$ as the respective rates. The first task is generated using intervals of $\upsilon_1 = 1$ units, whereas the second task is generated using intervals of $\upsilon_2 = 2$ units. All the inputs are uniformly distributed in the range $[0, 250]$. We generated 250 observations for task 1 and 125 for task 2. For training the multi-task model, we select $N_1 = 200$ from the 250 observations for task 1 and use all $N_2 = 125$ for the second task. The other 50 data points for task 1 correspond to a gap in the interval $[130, 180]$ that we use as the test set. In this experiment, we evaluated our model's capability in predicting one task, sampled more frequently, using the training information from a second task with a larger support.

In Figure 1 we show that the data in the second task, with a larger support, helps predicting the test data in the gap present in the first task, with a smaller support (right panel). However, this is not the case in the single task learning scenario where the predictions are basically constant and equal to 1 (left panel). Both models predict the training data equally well. SMSE (Standardized Mean Squared

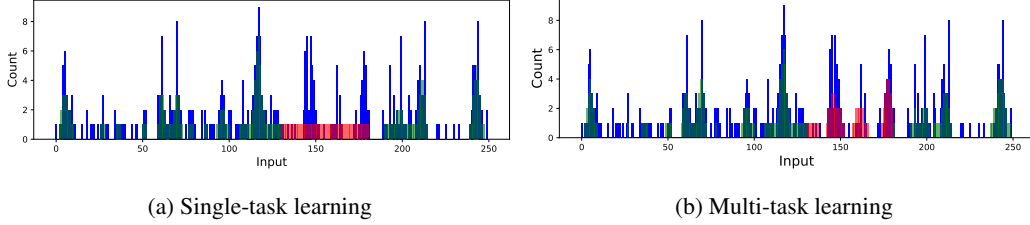

(a) Single-task learning          (b) Multi-task learning

Figure 1: Counts for the Poisson likelihoods and predictions using the single-task vs multi-task models. Predictions are shown only for the first task (the one with support of $v_1 = 1$). The blue bars are the original one-unit support data, the green bars are the predicted training count data and the red bars are the predicted test results in the gap $[130, 180]$. We did not include the two-unit support data (the second task) for clarity in the visualisation. The multi-task figure on the right (b) is illustrated again in Appendix A.4, Figure 7 for better visualisation.

Error) and SNLP (standardized negative log probability density) are calculated for five independent runs. For the multi-task scenario they are $0.464 \pm 0.136$ and $-0.822 \pm 0.109$ and for the single task case they are $0.9699 \pm 0.016$ and $-0.095 \pm 0.036$, respectively.

**Fertility rates from a Canadian census**   In this experiment, a subset of the Canadian fertility dataset is used from the Human Fertility Database (HFD) [1]. The dataset consists of live births' statistics by year, age of mother and birth order. The ages of the mother are between $[15, 54]$ and the years are between $[1944, 2009]$. It contains 2640 data points of fertility rate per birth order (the output variable) and there are four birth orders. We used the 2640 data points of the 1st birth only. The dataset was randomly split into 1640 training points and 1000 test points. We consider two tasks: the first task consists of a different number of data observations randomly taken from the 1640 training points. The second task consists of all the training data aggregated at two different resolutions, $5 \times 5$ and $2 \times 2$ (we wanted to test the predictive performance when the relation of high-resolution data to low-resolution data was $1^2$ to $5^2$ and another for $1^2$ to $2^2$). The aggregated data for the $5 \times 5$ case (a squared support of 5 years for the input `age` times 5 years for the input `years` of the study) is reduced to 104 data points and the aggregated data for $2 \times 2$ case is reduced to 660 points.

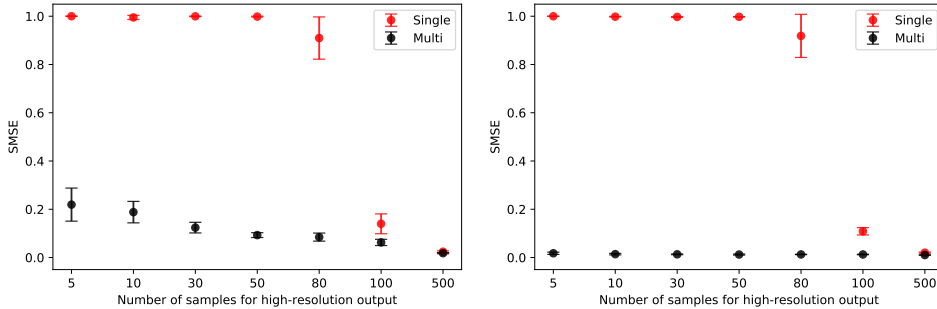

Figure 2: SMSE plots for the fertility dataset for $5 \times 5$ (left panel) and $2 \times 2$ (right panel) aggregated data. The Figure shows the performance in terms of the number of training instances used for the data sampled at a higher resolution. The test set always contains 1000 instances. We plot the mean and standard deviation for five repetitions of the experiment with different sets of training and test data. Appendix A.2 shows the same plots for SNLP. Appendix A.3 illustrates other experimental baselines that are compared to eachother for the same metrics. Further experiments considering more tasks are also included in Appendix A.4.

In the experiments, we train this multi-task model by slowly increasing the original resolution training data, while maintaining a fixed amount of training points mentioned before for the aggregated second task. The output variable (fertility rate for the first birth) was assumed to be Gaussian, so both tasks follow a Gaussian likelihood. We use $Q = 1$ with an EQ kernel $k_1(\mathbf{z}, \mathbf{z}')$ with $\mathbf{z} \in \mathbb{R}^2$ where

the two input variables are age of mother and birth year. We used 100 fixed inducing variables and mini-batches of size 50 samples. The prediction task consists of predicting the 1000 original resolution test data with the help of the second task which consists of the aggregated data ($5 \times 5$ or $2 \times 2$ for two separate experiments).

Figure 2 shows SMSE for five random selections of data points in the training and test sets. We notice that the multi-task learning model outperforms the single-task GP when there are few observations in the task with the original resolution data. This pattern holds below 500 observations. At that point, both models perform equally well since the single-task GP now has enough training data. With respect to the two different resolutions, the performance of the multi-task model is better when the second task has a $2 \times 2$ resolution rather than $5 \times 5$ resolution, as one might also expect.

**Air pollution monitoring network**    Particulate air pollution can be measured accurately with high temporal precision by using a $\beta$ attenuation (BAM) sensor or similar. Unfortunately these are often prohibitively expensive. We propose instead that one can combine the measurements from a low-cost optical particle counter (OPC) which gives good temporal resolution but is often badly biased, with the results of a Cascade Impactors (CIs), which are a cheaper method for assessing the mass of particulate species but integrate over many hours (e.g. 6 or 24 hours).

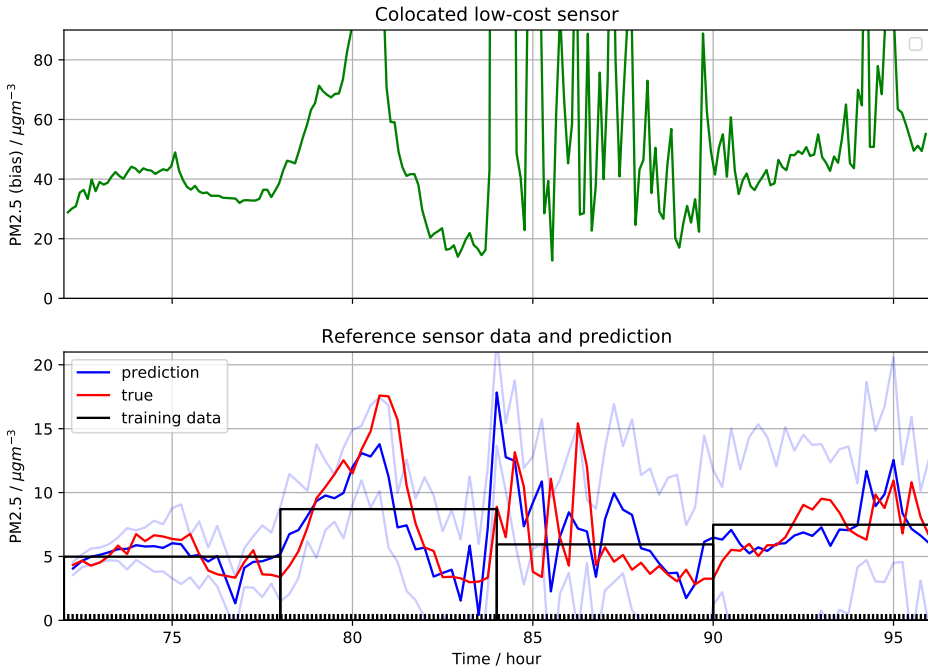

Figure 3: Upper plot: a (biased) OPC low-accuracy high-frequency measurement of PM2.5 air pollution. Lower plot: the high-precision low-frequency training data (black rectangles) the test data from the same instrument (red) and the posterior prediction for this output variable, predicting over the same 15-minute periods as the test data (blue, with pale blue indicating 95% confidence intervals). The ticks in the bottom of the lower plot indicate the position of the inducing inputs. Also, we have deliberately cut the higher peaks of the samples in the upper plot that can go as high as 500 $\mu g/m^3$, just to be able to visualise better the samples in other parts of the plot.

One can formulate the problem as observations of integrals of a latent function. The CI integrating over 6 hour periods while the OPC sensor integrating over short 5 minute periods. We used data from two fine particulate matter (PM) sensors. The sensors are less than 2.5 micrometer diameter (PM2.5) and are colocated in Kampala, Uganda at 0.3073°N 32.6205°E. The data is taken between 2019-03-13 and 2019-03-22. We used the average of six-hour periods from a calibrated mcerts-verified Osiris (Turnkey) particulate air pollution monitoring system as the low-resolution data, and then compared the prediction results to the original measurements (available at a 15 minute resolution). We used a PMS 5003 (Plantower) low-cost OPC to provide the high-resolution data. Typically we found these

values would often be biased. We simply normalised (scaled) the data to emphasise that the *absolute* values of these variables are not of interest in this model.

Our multi-task model consists of a single latent function, $Q = 1$, with covariance $k_1(z, z')$ that follows an EQ form. We assume both outputs follow Gaussian likelihoods. In our model, one task represents the high accuracy low-resolution samples and the second task represents the low-accuracy high-resolution samples. The posterior GP both aims to fulfil the 6-hour long integrals of the high-accuracy data (from the Osiris instrument) while remaining correlated with the high-frequency bias data from the OPC. We used 2000 iterations of the variational EM algorithm, with 200 evenly spaced inducing points and a fixed lengthscale of 0.75 hours. We only optimise the parameters of the coregionalisation matrix $\mathbf{B}_1 \in \mathbb{R}^{2 \times 2}$ and the variance of the noise of each Gaussian likelihood.

Figure 3 illustrates the results for a 24 hour period. The training data consists of the high-resolution low-accuracy sensor and a low-frequency high accuracy sensor. The aim is to reconstruct the underlying level of pollution both sensors are measuring. To test whether the additional high-frequency data improves the accuracy we ran the coregionalisation both with and without this additional training data.

We found that the SMSE for the predictions over the 9 days tested were substantially smaller with multi-task learning compared to using only the low-resolution samples, $0.439 \pm 0.114$ and $0.657 \pm 0.100$ respectively (the difference is statistical significant using a paired $t$-test with a $p$ value of 0.0008). In summary, the model was able to incorporate this additional data and use it to improve the estimates while still ensuring the long integrals were largely satisfied.

## 5   Conclusion

In this paper, we have introduced a powerful framework for working with aggregated datasets that allows the user to combine observations from disparate data types, with varied support. This allows us to produce both finely resolved and accurate predictions by using the accuracy of low-resolution data and the fidelity of high-resolution side-information. We chose our inducing points to lie in the latent space, a distinction which allows us to estimate multiple tasks with different likelihoods. SVI and variational-EM with mini-batches make the framework scalable and tractable for potentially very large problems. A potential extension would be to consider how the "mixing" achieved through coregionalisation could vary across the domain by extending, for example, the Gaussian Process Regression Network model [Wilson et al., 2012] to be able to deal with aggregated data. Such model would allow latent functions of different lengthscales to be relevant at different locations in the domain. In summary, this framework provides a vital toolkit, allowing a mixture of likelihoods, kernels and tasks and paves the way to the analysis of a very common and widely used data structure - that of values over a variety of supports on the domain.

## 6   Acknowledgement

MTS and MAA have been financed by the Engineering and Physical Research Council (EPSRC) Research Project EP/N014162/1. MAA has also been financed by the EPSRC Research Project EP/R034303/1.

## Footnotes

[1] https://www.humanfertility.org

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
