[Supplementary Material · multitask_aggregated_supplementary.pdf]

# Multi-task Learning for Aggregated Data using Gaussian Processes: Appendix

**Fariba Yousefi**        **Michael Thomas Smith**        **Mauricio A. Álvarez**

Department of Computer Science, University of Sheffield
{f.yousefi, m.t.smith, mauricio.alvarez}@sheffield.ac.uk

## A    Supplemental material

### A.1    Additional details on SVI

**Lower-bound**    It can be shown [Moreno-Muñoz et al., 2018] that the bound is given as

$$\mathcal{L} = \sum_{d=1}^{D} \sum_{j=1}^{N_d} \mathbb{E}\left[\log p(y_d(v_{d,j})|f_d(v_{d,j}))\right] - \mathrm{KL}(q(\mathbf{u})\|p(\mathbf{u})),$$

where the expected value is taken with respect to the $q(\mathbf{f}) = \int q(\mathbf{f}, \mathbf{u})d\mathbf{u}$ distribution, which is a Gaussian distribution with mean $\mathbf{K_{fu}K_{uu}^{-1}}\boldsymbol{\mu}$ and covariance $\mathbf{K_{ff}} + \mathbf{K_{fu}K_{uu}^{-1}(S - K_{uu})K_{uu}^{-1}K_{fu}^{\top}}$. For Gaussian likelihoods,

$$p(y_d(v_{d,j})|f_d(v_{d,j})) = \mathcal{N}(y_d(v_{d,j})|f_d(v_{d,j}), \sigma_{y_d}^2),$$

we can compute the expected value in the bound in closed form. For other likelihoods, we can use numerical integration to approximate it such as Gaussian-Hermite quadratures as in Hensman et al. [2015] and Saul et al. [2016]. Instead of using the whole batch of data $N = \sum_{d=1}^{D} N_d$, we can use mini-batches to estimate the gradient.

**Predictive distribution**    As we mentioned before, we are usually interested in the mean prediction $\mathbb{E}[\mathbf{y}_*]$ and the predictive variance $\mathrm{var}[\mathbf{y}_*]$. Both can be computed by exchanging integrals in the double integration over $\mathbf{y}_*$ and $\mathbf{f}_*$ For example, $\mathbb{E}[\mathbf{y}_*] = \int_{\mathbf{y}_*} \mathbf{y}_* p(\mathbf{y}_*|\mathbf{y}, \boldsymbol{\Upsilon}_*)d\mathbf{y}_* = \int_{\mathbf{f}_*} \int_{\mathbf{y}_*} \mathbf{y}_* p(\mathbf{y}_*|\mathbf{f}_*)d\mathbf{y}_* q(\mathbf{f}_*)d\mathbf{f}_*$. The inner integral in $\mathbb{E}[\mathbf{y}_*]$ is computed with the conditional distribution $p(\mathbf{y}_*|\mathbf{f}_*)$ and its form depend on the likelihood term per task. The outer integral can be approximated using numerical integration or Monte-Carlo sampling. A similar procedure can be followed to compute $\mathrm{var}[\mathbf{y}_*]$.

### A.2    SNLP for the fertility dataset

Figure 1 shows the results in terms of SNLP for the Fertility dataset. We can notice a similar pattern to the one observed for the SMSE in Figure 2 of the main paper.

Figure 1: SNLP for $5 \times 5$ and $2 \times 2$ aggregated data

## A.3 Comparing the model with different baselines for the fertility dataset

In Figure 2 different baselines are compared to the proposed method. Dependent GPs (DGP) [Boyle and Frean, 2005] and Intrinsic Co-regionalisation Model or Multi-task GPs (ICM) [Bonilla et al., 2008] use the centroid of the area as input. MTGPA (proposed method) performs better or similar to baselines as we increase the number of training points for the high-resolution output.

Figure 2: SMSE and SNLP plots for the fertility dataset for $5 \times 5$ (left panel) and $2 \times 2$ (right panel) aggregated data for different baselines, MTGPA , Independent GP (IND), DGP and ICM.

## A.4 Experimental results considering more tasks for the fertility dataset

In Figure 3 SNLP is calculated for four outputs (two outputs with high-resolution and a few data points and two outputs with low-resolution and many more data points). The high-resolution data correspond to the fertility rates of the first and second birth orders. The first task consists of a different number of data observations randomly taken from the training points of the fertility rate of the first birth. The second task consists of all the training data at the first task aggregated at two different resolutions, $5 \times 5$ and $2 \times 2$. The third task consists of a different number of data observations randomly taken from the training points of the fertility rate of the second birth. The fourth task consists of all the training data at the third task aggregated at two different resolutions, $5 \times 5$ and $2 \times 2$.

We use two different versions of our model and compare their SNLPs. In one version, all the outputs are considered as Gaussians (MTGPA) and in the second version, all the outputs are considered as heteroscedastic Gaussians (HetGPA). In the Gaussian case, only the mean parameter is modelled as a latent function, while the variance is a hyperparameter. However, in the heteroscedastic case, both mean and variance are assumed to follow latent functions. The model with HetGPA outperforms the model with MTGPA since it allows more flexibility toward the latent function that models the variance of the Gaussian.

Figure 3: SNLP plots for the fertility dataset for $5 \times 5$ (left panel) and $2 \times 2$ (right panel) aggregated data for four outputs (two fertility rates). All outputs are considered as Gaussian (MTGPA) and all outputs are considered as heteroscedastic Gaussian (HetGPA).

### A.5 Bar plot for the synthetic dataset

Figure 4 shows the result for the synthetic count data with Poisson likelihood and prediction using the multi-task model. This plot is the same as Figure 1.b of the main paper, however, the green bars are removed for better visualisation purposes.

Figure 4: Counts for the Poisson likelihood and predictions for the multi-task model. Predictions are illustrated only for the first task (the one with support of $\upsilon_1 = 1$). The blue bars are the original one-unit support data, and the red bars are the predicted test results in the gap $[130, 180]$. We did not include the two-unit support data (the second task) for clarity in the visualisation.