[Reviews · NeurIPS 2019]

Reviewer 1



ORIGINALITY: The authors present a framework for multi-task learning on aggregated data, i.e. data that has been averaged over time and/or space. The key contribution is to consider a *multi-task setting*, in which correlated aggregated data is jointly modeled and each task can have a different likelihood function. As such, the paper seems to be a relatively straightforward extension of the work of Moreno-Munoz er al. 2018 (Multi-output GPs when outputs have different likelihoods) by the work of Smith et al. 2018 (GPs for aggregated data in the single-output case). This combination is (to my knowledge) novel and relevant. QUALITY: Technically, the method seems sound and supported by the experiments, although the latter are partially presented in a manner that doesn't make them easy to follow. There is a clear setting when this method should be used (different kinds of correlated, aggregated data), but there is no assessment of weaknesses of the method. CLARITY: The paper is well organized, partially well written and easy to follow, in other parts with quite some potential for improvement, specifically in the experiments section. Suggestions for more clarity below. SIGNIFICANCE: I consider the work significant, because there might be many settings in which integrated data about the same quantity (or related quantities) may come at different cost. There is no earlier method that allows to take several sources of data into account, and even though it is a fairly straightforward extension of multi-task models and inference on aggregated data, it is relevant. MORE DETAILED COMMENTS: --INTRO & RELATED WORK: * Could you state somewhere early in the introduction that by "task" you mean "output"? * Regarding the 3rd paragraph of the introduction and the related work section: They read unnaturally separated. The paragraph in the introduction reads very technical and it would be great if the authors could put more emphasis there in how their work differs from previous work and introduce just the main concepts (e.g. in what way multi-task learning differs from multiple instance learning). Much of the more technical assessment could go into the related work section (or partially be condensed). --SECTION 2.3: Section 2 was straightforward to follow up to 2.3 (SVI). From there on, it would be helpful if a bit more explanation was available (at the expense of parts of the related work section, for example). More concretely: * l.145ff: $N_d$ is not defined. It would be good to state explicitely that there could be a different number of observations per task. * l.145ff: The notation has confused me when first reading, e.g. $\mathbb{y}$ has been used in l.132 for a data vector with one observation per task, and in l.145 for the collection of all observations. I am aware that the setting (multi-task, multiple supports, different number of observations per task) is inherently complex, but it would help to better guide the reader through this by adding some more explanation and changing notation. Also l.155: do you mean the process f as in l.126 or do you refer to the object introduced in l.147? * l.150ff: How are the inducing inputs Z chosen? Is there any effect of the integration on the choice of inducing inputs? l.170: What is z' here? Is that where the inducing inputs go? * l.166ff: It would be very helpful for the reader to be reminded of the dimensions of the matrices involved. * l.174 Could you explicitly state the computational complexity? * Could you comment on the performance of this approximate inference scheme based on inducing inputs and SVI? --EXPERIMENTS: * synthetic data: Could you give an example what kind of data could look like this? In Figure 1, what is meant by "support data" and what by "predicted training count data"? Could you write down the model used here explicitly, e.g. add it to the appendix? * Fertility rates: - It is unclear to me how the training data is aggregated and over which inputs, i.e. what you mean by 5x5. - Now that the likelihood is Gaussian, why not go for exact inference? * Sensor network: - l.283/4 You might want to emphasize here that CI give high accuracy but low time resolution results, e.g. "...a cheaper method for __accurately__ assessing the mass..." - Again, given a Gaussian likelihood, why do you use inducing inputs? What is the trade-off (computational and quality) between using the full model and SVI? - l.304ff: What do you mean by "additional training data"? - Figure 3: I don't understand the red line: Where does the test data come from? Do you have a ground truth? - Now the sensors are co-located. Ideally, you would want to have more low-cost sensors that high-cost (high accuracy) sensors in different locations. Do you have a thought on how you would account for spatial distribution of sensors? --REFERENCES: * please make the style of your references consistent, and start with the last name. Typos etc: ------------- * l.25 types of datasets * l.113 should be $f_{d'}(v')$, i.e. $d'$ instead of $d$ * l.282 "... but are badly bias" should be "is(?) badly biased" (does the verb refer to measurement or the sensor? Maybe rephrase.) * l.292 biased * Figure 3: biased, higher peaks, 500 with unit. * l.285 consisting of? Or just "...as observations of integrals" * l.293 these variables

Reviewer 2



-- Paper Summary -- Multi-output/multi-task Gaussian processes are highly suitable for several engineering and physical problems where multiple signals are associated with a given input. Although this problem has been thoroughly investigated in set-ups where every task or output is dependent on the same input or input space, it is also possible for related tasks to have different supports, where only aggregate data is available instead. In this work, a flexible model is developed for handling such data whereby each task can be assigned a different likelihood, and the input for each task can be of a different form or have different resolution. The effectiveness of this method is showcased using a synthetic example and two real-world problems. -- Writing/Clarity -- The paper is well-written - the broad motivation for developing the proposed model is clearly expressed in the opening sections of the paper, while the differences to similar work are succinctly highlighted in both the introduction and related work sections. I appreciated having a dual discussion of related work both before and after presenting the model since the contributions were contextualised better in this manner. I do however think that visualising some of the information in the form of graphical models (for depicting the model) and tables (for showing which features appear in related papers vs. this model - for example using ticks for model features/capabilities) could further clarify the sometimes overly-dense discussion. Further miscellaneous comments: - Spotted typos: L3: city, region or countries -> ‘city, region, or country’ or ‘cities, regions, or countries’; L15/122: this is -> that is/i.e.; L24: these types -> this category; L67: single task setting, we .. -> single task setting. We then…; L71/222: ‘Geostatistics’ doesn’t need to be capitalised; L74: mean zero -> zero mean; L91: incorrect punctuation in formula; L111: weight -> weigh; L124: has not a -> does not have a; L282/292/F3 caption: bias -> biased; L285: consisting ‘of’ observations; L290: available a at -> available at a; L313: demonstrated -> introduced; - Is there a citation for the variational EM procedure introduced in L183? - The bars in Figure 1 are difficult to interpret properly even when printed in colour; - Some capitalisation is required in the references, e.g. Bayesian, etc. Some references include links whereas others don’t - try to commit to a single format; -- Originality and Significance -- Due to the applied setting in which the problems investigated in this work typically arise, variations of this problem have been investigated in several other works, most recently by Law et al. (2018). There are several similar elements to this work, such as the use of stochastic variational inference for scalability and parameter optimisation, but the aforementioned work was primarily focused on count problems having a Poisson likelihood. This submission also places a greater emphasis on multi-task learning, while leveraging the methodology developed in Moreno-Muñoz et al (2018) for obtaining a coregionalisation scheme and heterogeneous predictions for different tasks. Parallels are also drawn to multiple instance learning, although the flexibility of using different likelihoods per task extends beyond the standard regression and classification problems considered in those problems. In view of the above, I believe that the paper succeeds in proposing and developing a very flexible framework for such problems, which generalises previous disparate work targeting a similar goal. The inclusion of a sensible inducing points framework also permits scalability to large problems, which further reinforces this work’s appeal for application to practical problems. Perhaps one of my complaints in this regard is that the problems chosen in the experiments are fairly conservative in all aspects regarding size, choice of likelihood, and number of tasks. While I appreciate that limiting the analysis to two tasks allows for easier visualisation of the results, I would have liked to see more difficult problems being considered. I am also slightly concerned with the assumption that ‘the correlation between tasks will remain constant over the whole domain’ [L320]. To the best of my understanding, this is very rarely the case in practice, where it is instead highly likely for the degree of correlation to be dependent on the location in the input-space. Given how the authors claim that this should only involve a minor extension of the currently presented model, I would highly encourage the authors to include this extension within the paper itself - this could then be investigated further through additional synthetic experiments. Given that this methodology is amenable to data with varying quality available at different resolutions/granularity, I would also expect some connection to be made to multi-fidelity modelling, which similarly leverages the availability of abundant lower fidelity data to improve high-fidelity predictions for which data is more scarce. Although I can immediately see the difference between the two set-ups in many aspects, e.g. lack of explicit ordering between tasks, varying support, etc, I believe there is still an interesting discussion to be made here with reference to basic multi-fidelity modelling and more recent state-of-the-art variations. -- Technical Quality/Evaluation -- The formulation of the model appears to be correct, and I did not spot any particular issues while going through the paper. Given that other competing models are not as flexible as what is being proposed here, the evaluation is consequently quite one-sided whereby the model is mostly considered against a very plain baseline. In this sense, the evaluation tends to seem more like a ‘demo’ at times. Nonetheless, the inclusion of a statistical test for verifying the contribution of the proposed model towards the observed results is a nice touch. However, I think it should still be possible to compare against the framework of Law et al (2018) in an experiment dealing with count data? It would be nice to have some details on the availability of the code - will this be implemented as an extension to GPy, GPflow or some other widely-used library? -- Overall recommendation -- I think this a solid paper overall - the work encompasses several recent developments in the literature on multi-task problems, and the proposed model has lots of potential use-cases due to its flexibility, even if these are currently not extensively showcased in the experiments. Nevertheless, I believe this paper should be considered for publication if the authors commit to carrying out further refinements and improvements. P.S. A paper having similar goals appeared on arXiv shortly following the NeurIPS deadline: “Multi-resolution Multi-task Gaussian Processes”, by Hamelijnck et al. In order to judge this submission fairly, I ignored the aforementioned paper when evaluating this work. Optionally, the authors may choose to briefly comment on the similarities/differences to this work. ** Post-rebuttal update ** I thank the authors for preparing a detailed rebuttal. On the topic of input-dependent coregionalisation, I would possibly opt for a slightly expanded discussion of the comment provided in the rebuttal rather than discarding its mention entirely. While not essential to the contribution developed in the paper, it remains a logical extension to the work, which is why some form of discussion would be appreciated. Alongside the new experiment summarised in the rebuttal, I also highly encourage the authors to identify more examples which showcase the flexibility of the model as described in the text. While I appreciate that it could be difficult to identify real-world problems which simultaneously showcase all of the model’s capabilities, this concern was also echoed in another review, which is why I believe it should be given due attention. Improving the overall presentation of the paper based on other suggestions provided in the reviews should also simplify the more the cluttered segments of the paper, and I look forward to reading an updated version which implements all of the aforementioned changes.

Reviewer 3



This paper proposes a general framework based on GPs for multi-task learning on the data aggregated over supports of different shapes and sizes. The change of support addressed herein is an important problem in various disciplines (e.g., geostatistics and epidemiology). The authors define the covariance function between any pair of supports as the double integration of the GPs, in which dependences between tasks are designed by the linear model of latent GPs. The inference procedure is based on variational EM that incorporates inducing points. The problem addressed in this submission is important, and the proposed approach is reasonable. However, there are several concerns; especially the experimental results seem not enough to support the authors' claims. Comments following the guidelines as requested. [Originality] Strengths. The main technical contribution of this submission is to extend the multi-task GP to handle the change of support; its idea sounds good and useful. Weaknesses. Some important related works are not discussed. Multi-task (i.e., multivariate) GPs have been widely studied in machine learning community. Although most of them assume that data values are associated with points, it would be better to mention several related multi-task GPs (e.g., [1],[2],[3]). Especially, [1] designed the dependent GP by a linear mixing of latent GPs, which is similar to this submission. Also, there is an important related work missing here: [4]. I think this paper essentially addressed a related task: Predicting the fine-grained data by using auxiliary data sets with various granularities. I would like the authors to clarify the differences and advantages of this submission. [Quality] Strengths. This paper is technically sound except for some concerns. The authors evaluate the proposed model in the simple experimental setting using synthetic and real data sets. Weaknesses. My concerns about the proposed model are as follows: 1) I have understood that the integral in Equation (1) corresponds to bag observation model in [Law et al., NeurIPS'18] or spatial aggregation process in [4]. The formulation introduced by the authors assume that the observations are obtained by averaging over the corresponding support $v$. However, the data might be aggregated by another procedure, e.g., simple summation or population weighted average; actually the disease incident data are often available in count, or rate per the number of residents. 2) In order to handle various data types (e.g., count and rate), shouldn't the corresponding aggregation processes be performed at the likelihood level? 3) I think it would be more efficient to estimate ${a_{d,q}}$ instead of $B_q$ since $b^q_{d,d'} = a_{d,q}a_{d',q}$. The major weakness of this submission is in the experiments. First, the proposed model should be compared with any typical baseline, such as regression-based model with aggregation process (e.g., Law et al., NeurIPS'18, [4]) and multi-task GP with point-referenced data (e.g., [1]). I believe the previous multi-task GP can be applied via the simplification; that is, each data value at the support $v$ is assumed to be associated with the representative point (e.g., centroid) of its support (as in the previous work [4]). Second, the extensive experiments are helpful to verify the effectiveness of the proposed model. In all the experiments, the authors consider two tasks. I would like to see the experimental results considering more tasks; then it is a good idea to discuss how to determine the number of latent GPs $Q$. Short question: I was wondering if you could give me the detail of *resolution 5 \times 5* in the experimental setting of fertility rates. [Clarity] This paper is easy to understand. Some typos: 1) In line 235, *low-cost* should be *low-accuracy*? 2) In line 239, *GP process* should be *GP*. [Significance] Aggregated data with different supports are commonplace in a wide variety of applications, so I think this is an important problem to tackle. However, the major weakness of the submission in my view is that the evaluation of the proposed model is not enough, so the effectiveness/usefulness of the model is unclear from the experimental results. I think it would be great to compare the proposed model with baseline methods. [1] Y. W. Teh et al., Semiparametric Latent Factor Models, AISTATS, 333-340, 2005. [2] P. Boyle et al., Dependent Gaussian Processes, NeurIPS, 217-224, 2005. [3] E. Bonilla et al., Multi-task Gaussian Process Prediction, NeurIPS, 153-160, 2008. [4] Y. Tanaka et al., Refining Coarse-grained Spatial Data Using Auxiliary Spatial Data Sets with Various Granularities, AAAI, 2019. https://arxiv.org/abs/1809.07952 ------------------------------ After author feedback: I appreciate the responses to my questions. The new experimental results in the rebuttal is a welcome addition. In light of this, I upgraded my score. The proposal is a combination of the coregionalization and the concept of aggregation process used in block-kriging; this is a simple but effective way. I also agree that a sensor experiment is one of the applications with the proposed model. But I'm still of the opinion that there is not enough experiments and/or discussions to support the authors' claims. The authors state that the model is a general framework and has many applications related to geostatistics (lines 14-23); the support $v$ corresponds to the 2-dimensional region, e.g., borough (line 92). As described in Related work (lines 222-229), the proposed model strongly relates to spatial downscaling and disaggregation in geostatistics. If anything, I think this application that contains spatial aggregation is a more critical one for the proposed model. In the spatial data setting, a wide variety of data sets is available at various spatial granularities (for instance, New York City publish open data in [https://opendata.cityofnewyork.us]). Naturally, one would like to handle these data sets simultaneously (as in Law et al., NeurIPS'18, [4]); namely the setting with a large number of tasks. In that case, I believe the authors should discuss several issues; for example, the sensitivity of the number of latent GPs $Q$, the approximate accuracy of integral over regions, etc. I think it would be better to clarify the scope of this study and discuss the above issues.

[Author Response · NeurIPS 2019]

We thank all reviewers for their useful comments and positive feedback. We'll fix all minor comments and typos. **Reviewer (R) 1** *"The paragraph in the introduction reads very technical (...)"* In the Introduction, we focus on GP models for aggregated data and multiple instance learning whereas the Related work section mentions work beyond GPs. At the time of submission, we didn't know of any other multi-task GP model for aggregated data. There are two recent submissions to arxiv (see reply to R2) that we'll use to follow the reviewer's recommendation. *"Section 2 was straightforward to follow up to 2.3 (SVI). (...)"* Due to space restrictions we summarised the theory from Section 2.3 since this piece of literature has been described before in Moreno-Muñoz et al. (2018). We'll expand the description of Section 2.3 in the Appendix. *"l.145ff: The notation has confused me when first reading, (...)"* In L132 we use $\mathbf{y}(v)$ to refer to the vector of outputs as a function of $v$ and in L145 $\mathbf{y}$, without the argument $v$, to refer to the output vector of the dataset. We'll clarify in the paper. *"Also l.155: do you mean the process $f$ (...)"* Yes, we refer to $f$ *"l.150ff: How are the inducing inputs Z chosen? (...)"* We use $k$-means over the input data with $k = M$, the number of inducing inputs. We fix them during optimisation and assume the inducing inputs are points, but we could have also defined them as intervals or supports. *"(...) computational complexity?"* Similar to the one in Moreno-Muñoz et al. (2018): $\mathcal{O}(QM^3 + JNQM^2)$, where $J$ depends on the type of likelihood. *"synthetic data: Could you give an example (...)?"* These could represent two histograms, for example, defined over bins with different sizes. *"what is meant by "support data" (...)"* One-unit support data: data with a support of one unit. *"predicted training count data""* Predictions made by the trained model over the training data. *"what you mean by 5x5"* a squared support of 5 years for the input `age` times 5 years for the input `years` of the study. *"Now that the likelihood is Gaussian, why not go for exact inference"* That's true, but this wouldn't work in the general case, for example, we couldn't apply this for the toy example. *"Figure 3: I don't understand the red line:"* It is the ground truth obtained directly by the sensor. We'll clarify this in the new version. *"Do you have a thought (...) sensors (...)"* An idea previously used in other papers is to assume that each spatial location is a different output. We're looking into this for our application in air pollution. *"Extend explanation in Section 2.3 (...)"* *"Rewrite the section on experiments (...)."* We'll do this as explained above. **Reviewer 2** *"graphical models(..)."* We'll add graphical models to the final version. *"bars in Figure 1 (...)"* These are meant to be read as histograms. We'll add another plot zooming in the prediction range. *"(...) assumption that 'the correlation between tasks will remain constant (...)'"* Our most sincere apologies. This is in no way straightforward and will involve a model along the lines of the Gaussian process regression networks (Wilson et al, 2011). We will remove this statement from the manuscript. *"(...) availability of the code"* We have our code on GPy, and we'll make it available after the decision. *"P.S. A paper having similar goals appeared on arXiv (...)"* Two papers with similar goals appeared on arXiv recently, the one mentioned by the reviewer, "Multi-resolution Multi-task GPs" (arxiv1) and "Spatially Aggregated GPs with Multivariate Areal Outputs" (arxiv2). Differences: we allow heterogeneous likelihoods (compared to arxiv1 and arxiv2), an exact solution to the integration of the latent function through the kernel in Smith et al (2018) (different to arxiv1); and inducing inputs for computational complexity (different to arxiv2). We'll add these references. *"(...) discussion regarding multi-fidelity methods (...)"* Very relevant, thanks. We'll add this to the discussion. *"If possible, adding a more involved experiment (...)"* See reply to R3 (experiment on more tasks). **Reviewer 3** *" (...) mention several related multi-task GPs (e.g., [1],[2],[3])."* We'll add the references. [1] and [3] are particular cases of LMC as it has been described in detail by Alvarez et al (2012). *"(...) related work missing here: [4] (...) differences and advantages (...)"* [4] does not attempt to do simultaneous prediction of several variables, only one variable is considered. They mainly use GPs for creating data from different auxiliary sources. Other differences: they only consider Gaussian regression and they do not include inducing variables. *"the data might be aggregated by another procedure, e.g., simple summation or population weighted average; "* Agreed. Our motivation was to have a general purpose model. Other types of aggregation will require prior knowledge by the user. This can be extended in future work. *"(...) aggregation (...) at the likelihood level?"* It's happening in a sense because the latent functions obtained after the integration modulate the parameters of each likelihood in different ways, depending on $a_{d_q}$. *"(...) I think it would be more efficient to estimate $a_{d,q}$ instead of $B_q$."* We estimate $\mathbf{B}_q$ by estimating first the Cholesky factor $\mathbf{L}_q$. See L187. This is efficient. *"the proposed model should be compared with any typical baseline"* See Fig 1. We'll include the $5 \times 5$ resolution case and the SNLP metric in the new version and similar baselines for the other experiments (toy and air pollution). *"experimental results considering more tasks"* We ran another experiment with the Fertility dataset: four outputs (two high-res few data points, two low-res many more data points) and compared two versions of our model: all outputs as Gaussians and all outputs as heteroscedastic Gaussians. SMSEs for both models are comparable, but the model with heteroscedastic Gaussians outperforms in terms of the SNLP. We'll add this experiment to the new version with many more details. *"*resolution $5 \times 5$* (...)"* See reply to R1.

Figure 1: Fertility rates $2\times2$ low resolution case. MTGPA (our method); IND (Independent GP with aggregated inputs); DGP (Dependent GPs, ref [2], R3); ICM (Intrinsic Co-regionalisation Model or Multi-task GPs, ref [3], R3). DGP and ICM use the centroid of the area as input. MTGPA performs better or similar to baselines as we increase the number of training points for the high-res output.

[Meta-Review · NeurIPS 2019]

The reviewers reached a consensus on the technical merits of the paper. There is some agreement that further experiments would help, and the authors have stated that they will include another experiment. While I agree that spatial downscaling is a natural application of your model (and thus it would make sense to discuss this somewhere in the paper) I think it is fine to leave this particular application for future work.